# Entanglement Property of Tripartite GHZ State in Different Accelerating Observer Frames [note 1]

**DOI:** 10.3390/e24081011

**Published:** 2022-07-22

**Authors:** Qian Dong, Roberto de Jesus León-Montiel, Guo-Hua Sun, Shi-Hai Dong

**Affiliations:** 1Centro de Investigación en Computación, Instituto Politécnico Nacional, UPALM, Mexico City 07738, Mexico; dldongqian@gmail.com (Q.D.); sunghdb@yahoo.com (G.-H.S.); 2Instituto de Ciencias Nucleares, Universidad Nacional Autónoma de México, Apartado Postal 70-543, Mexico City 04510, Mexico; roberto.leon@nucleares.unam.mx; 3Research Center for Quantum Physics, Huzhou University, Huzhou 313000, China; 4Laboratorio de Información Cuántica, CIDETEC, Instituto Politécnico Nacional, UPALM, Mexico City 07700, Mexico

**Keywords:** single mode approximation, entanglement measures, Dirac field, noninertial frames, 03.67.a, 03.67.Mn, 03.65.Ud, 04.70.Dy

## Abstract

According to the single-mode approximation applied to two different mo des, each associated with different uniformly accelerating reference frames, we present analytical expression of the Minkowski states for both the ground and first excited states. Applying such an approximation, we study the entanglement property of Bell and Greenberger–Horne–Zeilinger (GHZ) states formed by such states. The corresponding entanglement properties are described by studying negativity and von Neumann entropy. The degree of entanglement will be degraded when the acceleration parameters increase. We find that the greater the number of particles in the entangled system, the more stable the system that is studied by the von Neumann entropy. The present results will be reduced to those in the case of the uniformly accelerating reference frame.

## 1. Introduction

Up to now, quantum entanglement has become the central resource for quantum information as a new and fast-growing field. Its development has been concerned with many useful branches, e.g., quantum computation, quantum cryptography and quantum teleportation. Quantum entanglement plays a fundamental part in the creation of a large amount of information protocols, such as the Quantum Key Distribution [1,2,3]. In recent years, a new, emerging field named relativistic quantum information, which combines quantum information theory, quantum field theory and general relativity, has been developed in quantum information. Its central question is the study of the entanglement measures in a noninertial frame. Previous studies show that the entanglement between modes of bosonic or fermionic fields is degraded from the perspective of observers moving in a uniform acceleration; therefore, we need to have a quantitative understanding of such degradation. Based on this idea, a single-mode approximation has been proposed and developed widely in relativistic quantum information [4,5,6,7,8,9,10,11,12,13,14,15,16,17,18,19,20,21,22,23,24]. For example, Asling and his coauthors have used a single-mode approximation to study the behavior of the entanglement between the modes of a free Dirac field in a noninertial frame in flat space-time from the point of view of two observers, Alice and Rob, in a relative uniform acceleration [9]. After that, the entanglement properties of bipartite and multipartite, including the tripartite, tetrapartite and pentapartite entangled systems, have been studied. These systems are concerned with the GHZ, generalized GHZ, W-class pure states and Werner mixed state, etc.

However, in the commonly used single-mode approximation, only one uniformly accelerated observer is concerned, and thus, there are some limitations in applications, so it still remains to consider the case in which two different uniformly accelerated observers exist. Similar to the single-mode approximation, we still want to obtain a quantitative expression and show how two different acceleration parameters $r_{j}$ and $r_{k}$ affect the entanglement property.

In this paper, based on the single-mode approximation, we derive analytical expressions of two different accelerating observer frames. In Section 2, we first review the transformation between Minkowski, Unruh and Rindler modes, and then we present an approximate transformation by considering different accelerations of the observer frames. We shall use this transformation to study the entanglement of the Bell and GHZ states in Section 3 through the calculation of their negativities and the von Neumann entropy. Finally, we present the conclusions in Section 4.

## 2. Generalization of Single Mode Approximation

Let us consider a free Minkowski Dirac field in the single mode approximation before we generalize it to the case for two different accelerating observer frames. The field ψ can be expanded in terms of the positive (fermions) and negative (antifermions) energy solutions of the Dirac equation $\psi_{k}^{+}$ and $\psi_{k}^{-}$ since they form a completely orthogonal set of modes [25,26,27,28]
(1)$$\psi =\int \left(a_{k}\psi_{k}^{+}+b_{k}^{†}\psi_{k}^{-}\right)\;dk,$$
where *k* is the notation for the wave vector *k*, and the positive and negative energy Minkowski modes have the form
(2)$${\left(\psi_{k}^{\pm }\right)}_{s}=\frac{1}{\sqrt{2\pi w_{k}}}\phi_{s}^{\pm }e^{\pm i\left(\mathrm{kx}-tw_{k}\right)},\;$$
where $w_{k}={\left(m^{2}+k^{2}\right)}^{1/2}$ and $\phi_{s}$ is a constant spinor with $s=\left\{↑,↓\right\}$, and all the wave functions satisfy the normalization relation. The operators $a_{k}^{†},b_{k}^{†}$ and $a_{k},b_{k}$ are the creation and annihilation operators for the positive and negative energy solutions of momentum *k*, which satisfy the anticommutation relations
(3)$$\left\{a_{i},a_{j}^{†}\right\}=\left\{b_{i},b_{j}^{†}\right\}=\delta_{ij}.\;$$

The definition of the Minkowski vacuum state in an inertial frame is
(4)$$|0\rangle =\prod_{k\;k^{\prime }}{|0_{k}\rangle }^{+}{|0_{k^{\prime }}\rangle }^{-},\;$$
where the signs $\left\{+,-\right\}$ are used to denote the particle and antiparticle vacua, so we have $a_{k}{|0_{k}\rangle }^{+}=b_{k}{|0_{k}\rangle }^{-}=0$, ${\left(a_{k}^{†}\right)}^{2}={\left(b_{k}^{†}\right)}^{2}=0$ and ${|1_{k}\rangle }^{+}=a_{k}^{†}{|0_{k}\rangle }^{+}$. To study the Bell and GHZ states in the potentially different accelerating observer frame, it is helpful for us to use Rindler coordinates and divide Minkowski space-time into two inaccessible regions I and II. For convenience, in this work, we denote the inertial observers Alice, Bob and Charlie as *A*, *B* and *C*, respectively. Following the pioneering work [9] and our recent study [28], we use the $\left\{c_{k}^{I,\mathrm{II}},c_{k}^{I†,\mathrm{II}†}\right\}$ to denote the annihilation and creation operators for fermions (particles) and $\left\{d_{k}^{I,\mathrm{II}},d_{k}^{I†,\mathrm{II}†}\right\}$ to denote the annihilation and creation operators for antifermions (antiparticles) in regions I and II, respectively, so that Equation (Equation 1) can be rewritten as
(5)$$\psi =\int \left(c_{k}^{I}\psi_{k}^{I+}+d_{k}^{I†}\psi_{k}^{I-}+c_{k}^{\mathrm{II}}\psi_{k}^{\mathrm{II}+}+d_{k}^{\mathrm{II}†}\psi_{k}^{\mathrm{II}-}\right)\;dk.$$

To avoid the repeated calculation, we suggest the reader refer to recent pioneering contributions to this topic [29,30,31,32,33,34,35]. Using the relation between the Minkowski and Rindler creation and annihilation operators satisfying the Bogoliubov transformation, we are able to obtain [9]
(6)$${|0_{k}\rangle }^{+}=\cos\left(r\right){|0_{k}\rangle }_{I}^{+}{|0_{-k}\rangle }_{\mathrm{II}}^{+}+e^{-i\phi }\sin\left(r\right){|1_{k}\rangle }_{I}^{+}{|1_{-k}\rangle }_{\mathrm{II}}^{+}={|0\rangle }_{R}$$
and
(7)$$a_{k}^{†}{|0\rangle }_{R}=c_{k}^{I†}{|0_{k}\rangle }_{I}^{+}{|0_{-k}\rangle }_{\mathrm{II}}^{-}={|1_{k}\rangle }_{I}^{+}{|0_{-k}\rangle }_{\mathrm{II}}^{-}={|1\rangle }_{R}.$$

Now, let us consider the case in which we consider a superposition of two annihilation operators on modes *j* and *k*, acting in region I, that is, $c_{jk}^{I}=\left(\omega_{j}c_{j}^{I}+\omega_{k}c_{k}^{I}\right)$, where two complex coefficients of this superposition given as $\omega_{j}$ and $\omega_{k}$, respectively, satisfy the normalization condition $|\omega_{k}{|}^{2}+{\left|\omega_{j}\right|}^{2}=1$. For a single mode *k*, based on the definition of the operator *S* [9],
(8)$$S=\exp[re^{-i\phi }c_{k}^{I†}d_{-k}^{\mathrm{II}†}+re^{i\phi }c_{k}^{I}d_{-k}^{\mathrm{II}}],$$
when applied to two different modes, each associated with different uniformly accelerating reference frames, we are able to express the operator $S_{jk}$ as follows:(9)$$\begin{array}{c} S_{jk}=S_{j}⊗S_{k}\\ \;\;\;\;\;\;=\exp\left\{r_{j}e^{-i\phi }c_{j}^{I†}d_{-j}^{\mathrm{II}†}+r_{j}e^{i\phi }c_{j}^{I†}d_{-j}^{\mathrm{II}†}\right.\\ \;\;\;\;\;\;\left.+r_{k}e^{-i\phi }c_{k}^{I†}d_{-k}^{\mathrm{II}†}+r_{k}e^{i\phi }c_{k}^{I†}d_{-k}^{\mathrm{II}†}\right\}. \end{array}$$

By using this relation, we can obtain
(10)$$\begin{array}{c} a_{jk}=S_{jk}c_{jk}^{I}S_{jk}^{†}\\ \;\;\;\;\;\;=\omega_{j}\left[\cos\left(r_{j}\right)c_{j}^{I}-e^{-i\phi_{j}}\sin\left(r_{j}\right)d_{j}^{\mathrm{II}†}\right]\\ \;\;\;\;\;\;+\omega_{k}\left[\cos\left(r_{k}\right)c_{k}^{I}-e^{-i\phi_{k}}\sin\left(r_{k}\right)d_{k}^{\mathrm{II}†}\right] \end{array}$$
and
(11)$$\begin{array}{c} b_{-jk}^{†}=S_{jk}d_{-jk}^{\mathrm{II}†}S_{jk}^{†}\\ \;\;\;\;\;\;\;={\bar{\omega }}_{j}\left[\cos\left(r_{j}\right)d_{j}^{\mathrm{II}†}+e^{i\phi_{j}}\sin\left(r_{j}\right)c_{j}^{I}\right]\\ \;\;\;\;\;\;\;+{\bar{\omega }}_{k}\left[\cos\left(r_{k}\right)d_{k}^{\mathrm{II}†}+e^{i\phi_{k}}\sin\left(r_{k}\right)c_{k}^{I}\right]. \end{array}$$

In a Minkowski vacuum space, $a_{jk}$ and $b_{-jk}$ annihilate the two-mode particle and antiparticle ($a_{jk}{|0_{jk}\rangle }^{+}=0$ and $b_{-jk}{|0_{jk}\rangle }^{+}=0$). For two different modes, each associated with different uniformly accelerating reference frames, we have
(12)$$\begin{array}{c} {|0_{jk}\rangle }^{+}={|0_{j}\rangle }^{+}⊗{|0_{k}\rangle }^{+}\\ \;\;\;\;\;\;\;\;\;=\left(\sum_{n=0}^{1}j_{n}{|n_{j}\rangle }_{I}^{+}{|n_{-j}\rangle }_{\mathrm{II}}^{-}\right)⊗\left(\sum_{n=0}^{1}k_{n}{|n_{k}\rangle }_{I}^{+}{|n_{-k}\rangle }_{\mathrm{II}}^{-}\right)\\ \;\;\;\;\;\;\;\;\;=j_{0}k_{0}|0_{\hat{jI}}0_{\hat{j\mathrm{II}}}0_{\hat{kI}}0_{\hat{k\mathrm{II}}}\rangle +j_{0}k_{1}|0_{\hat{jI}}0_{\hat{j\mathrm{II}}}1_{\hat{kI}}1_{\hat{k\mathrm{II}}}\rangle \\ \;\;\;\;\;\;\;\;\;\;+j_{1}k_{0}|1_{\hat{jI}}1_{\hat{j\mathrm{II}}}0_{\hat{kI}}0_{\hat{k\mathrm{II}}}\rangle +j_{1}k_{1}|1_{\hat{jI}}1_{\hat{j\mathrm{II}}}1_{\hat{kI}}1_{\hat{k\mathrm{II}}}\rangle . \end{array}$$
Based on Equations (Equation 10) and (Equation 12) we obtain $a_{jk}{|0_{jk}\rangle }^{+}=0$, i.e.,
(13)$$\begin{array}{c} 0=[\omega_{j}\left(\cos\left(r_{j}\right)c_{j}^{I}-e^{-i\phi_{j}}\sin\left(r_{j}\right)d_{j}^{\mathrm{II}†}\right)\\ \;\;\;\;\;\;+\omega_{k}\left(\cos\left(r_{k}\right)c_{k}^{I}-e^{-i\phi_{k}}\sin\left(r_{k}\right)d_{k}^{\mathrm{II}†}\right)]\\ \;\;\;\;\;\;[j_{0}k_{0}|0_{\hat{jI}}0_{\hat{j\mathrm{II}}}0_{\hat{kI}}0_{\hat{k\mathrm{II}}}\rangle +j_{0}k_{1}|0_{\hat{jI}}0_{\hat{j\mathrm{II}}}1_{\hat{kI}}1_{\hat{k\mathrm{II}}}\rangle \\ \;\;\;\;\;\;+j_{1}k_{0}|1_{\hat{jI}}1_{\hat{j\mathrm{II}}}0_{\hat{kI}}0_{\hat{k\mathrm{II}}}\rangle +j_{1}k_{1}|1_{\hat{jI}}1_{\hat{j\mathrm{II}}}1_{\hat{kI}}1_{\hat{k\mathrm{II}}}\rangle ]. \end{array}$$
After simplifying the equation
(14)$$\begin{array}{c} j_{1}\cos\left(r_{j}\right)-j_{0}e^{-i\phi_{j}}\sin\left(r_{j}\right)=0,\\ k_{1}\cos\left(r_{k}\right)-k_{0}e^{-i\phi_{k}}\sin\left(r_{k}\right)=0, \end{array}$$
we have the result
(15)$$\begin{array}{c} j_{1}=j_{0}e^{-i\phi_{j}}\frac{\sin(r_{j})}{\cos(r_{j})}=j_{0}e^{-i\phi_{j}}\tan\left(r_{j}\right),\\ k_{1}=k_{0}e^{-i\phi_{k}}\frac{\sin(r_{k})}{\cos(r_{k})}=k_{0}e^{-i\phi_{k}}\tan\left(r_{k}\right). \end{array}$$
By substituting them into Equation (Equation 12)
(16)$$\begin{array}{c} {|0_{jk}\rangle }^{+}=j_{0}k_{0}|0_{\hat{jI}}0_{\hat{j\mathrm{II}}}0_{\hat{kI}}0_{\hat{k\mathrm{II}}}\rangle \\ \;\;\;\;\;\;\;\;\;+j_{0}k_{0}e^{-i\phi_{j}}\tan\left(r_{j}\right)|1_{\hat{jI}}1_{\hat{j\mathrm{II}}}0_{\hat{kI}}0_{\hat{k\mathrm{II}}}\rangle \\ \;\;\;\;\;\;\;\;\;+j_{0}k_{0}e^{-i\phi_{j}}\tan\left(r_{k}\right)|0_{\hat{jI}}0_{\hat{j\mathrm{II}}}1_{\hat{kI}}1_{\hat{k\mathrm{II}}}\rangle \\ \;\;\;\;\;\;\;\;\;+j_{0}k_{0}e^{-i\phi_{j}-i\phi_{k}}\tan\left(r_{j}\right)\tan\left(r_{k}\right)|1_{\hat{jI}}1_{\hat{j\mathrm{II}}}1_{\hat{kI}}1_{\hat{k\mathrm{II}}}\rangle , \end{array}$$
and then normalizing the state
(17)$${\left({|0_{jk}\rangle }^{+}\right)}^{†}{|0_{jk}\rangle }^{+}=1,\;\;{\left(j_{0}k_{0}\right)}^{2}\sec^{2}\left(r_{j}\right)\sec^{2}\left(r_{k}\right)=1,$$
we have $j_{0}k_{0}=\pm \sqrt{\cos\left(r_{j}\right)\cos\left(r_{k}\right)}$. Substituting this into (Equation 16) allows us to find
(18)$$\begin{array}{c} {|0\rangle }_{R}={|0_{jk}\rangle }^{+}=\cos\left(r_{j}\right)\cos\left(r_{k}\right)|0_{\hat{jI}}0_{\hat{j\mathrm{II}}}0_{\hat{kI}}0_{\hat{k\mathrm{II}}}\rangle \\ \;\;\;\;\;\;\;\;\;\;\;\;\;\;\;\;\;\;+e^{-i\phi_{k}}\cos\left(r_{j}\right)\sin\left(r_{k}\right)|0_{\hat{jI}}0_{\hat{j\mathrm{II}}}1_{\hat{kI}}1_{\hat{k\mathrm{II}}}\rangle \\ \;\;\;\;\;\;\;\;\;\;\;\;\;\;\;\;\;\;+e^{-i\phi_{j}}\cos\left(r_{k}\right)\sin\left(r_{j}\right)|1_{\hat{jI}}1_{\hat{j\mathrm{II}}}0_{\hat{kI}}0_{\hat{k\mathrm{II}}}\rangle \\ \;\;\;\;\;\;\;\;\;\;\;\;\;\;\;\;\;\;+e^{-i\phi_{j}-i\phi_{k}}\sin\left(r_{j}\right)\sin\left(r_{k}\right)|1_{\hat{jI}}1_{\hat{j\mathrm{II}}}1_{\hat{kI}}1_{\hat{k\mathrm{II}}}\rangle . \end{array}$$

For the excited state, however, one has
(19)$${|1\rangle }_{R}={|1_{jk}\rangle }^{+}=a_{jk}^{†}{|0_{jk}\rangle }^{+}$$
where
(20)$$\begin{array}{c} a_{jk}^{†}=\omega_{j}^{*}\left(\cos\left(r_{j}\right)c_{j}^{†}-e^{i\phi_{j}}\sin\left(r_{j}\right)d_{j}\right)\\ \;\;\;\;\;+\omega_{k}^{*}\left(\cos\left(r_{k}\right)c_{k}^{†}-e^{i\phi_{k}}\sin\left(r_{k}\right)d_{k}\right). \end{array}$$

Using the following properties,
(21)$$\begin{array}{c} a_{jk}{|0_{jk}\rangle }^{+}=a_{jk}{|0\rangle }_{R}=0,\;\;a_{jk}^{†}{|0_{jk}\rangle }^{+}=a_{jk}^{†}{|0\rangle }_{R}={|1\rangle }_{R},\\ {\left(a_{jk}^{†}\right)}^{2}{|0_{jk}\rangle }^{+}={\left(a_{jk}^{†}\right)}^{2}{|0\rangle }_{R}=a_{jk}^{†}{|1\rangle }_{R}=0, \end{array}$$
we can finally obtain
(22)$$\begin{array}{c} {|1\rangle }_{R}={|1_{jk}\rangle }^{+}=\omega_{k}^{*}\cos\left(2r_{k}\right)\cos\left(r_{j}\right)|0_{\hat{jI}}0_{\hat{j\mathrm{II}}}1_{\hat{kI}}0_{\hat{k\mathrm{II}}}\rangle \\ \;\;\;\;\;\;\;\;\;\;\;\;\;\;\;\;\;\;+\omega_{j}^{*}\cos\left(2r_{j}\right)\cos\left(r_{k}\right)|1_{\hat{jI}}0_{\hat{j\mathrm{II}}}0_{\hat{kI}}0_{\hat{k\mathrm{II}}}\rangle \\ \;\;\;\;\;\;\;\;\;\;\;\;\;\;\;\;\;\;+e^{-i\phi_{k}}\omega_{j}^{*}\cos\left(2r_{j}\right)\sin\left(r_{k}\right)|1_{\hat{jI}}0_{\hat{j\mathrm{II}}}1_{\hat{kI}}1_{\hat{k\mathrm{II}}}\rangle \\ \;\;\;\;\;\;\;\;\;\;\;\;\;\;\;\;\;\;+e^{-i\phi_{j}}\omega_{k}^{*}\cos\left(2r_{k}\right)\sin\left(r_{j}\right)|1_{\hat{jI}}1_{\hat{j\mathrm{II}}}1_{\hat{kI}}0_{\hat{k\mathrm{II}}}\rangle . \end{array}$$

## 3. Fermionic Entanglement in Two Different Accelerating Observer Frames

When single mode approximation is applied to two different modes, each associated with different uniformly accelerating reference frames, we use *j* and *k* to represent the modes on states, but satisfy relation $|\omega_{k}{|}^{2}+{\left|\omega_{j}\right|}^{2}=1$. ${|n\rangle }_{A}$ means Alice for the Minkowski particle mode ${|n\rangle }^{+}$, ${|n\rangle }_{I}$ for the Rindler region I particle mode ${|n\rangle }_{I}^{+}$, and ${|n\rangle }_{\mathrm{II}}$ for the Rindler region II antiparticle mode ${|n\rangle }_{\mathrm{II}}^{-}$. Similarly, we simplify the description “Minkowski mode for Alice” to “mode A” and the Rindler particle and antiparticle modes in regions I and II to “mode I” and “mode II”, respectively. Before studying GHZ states, we first consider the Bell state in an inertial frame, $|\phi \rangle =\frac{1}{\sqrt{2}}\left(|0_{\hat{A}}0_{\hat{R}}\rangle +|1_{\hat{A}}1_{\hat{R}}\rangle \right)$. After expanding the Minkowski particle state into the Rindler region I and II (particle and antiparticle) and the mode *j* and *k* using Equations (Equation 18) and (Equation 22), we can obtain the following matrix form
(23)$$\begin{array}{c} \rho_{\mathrm{AjIkI}}=\frac{1}{2}\left(\begin{array}{cccccccc} \beta^{2}\delta^{2} & 0 & 0 & 0 & 0 & \beta^{2}\delta \eta_{2}\omega_{k} & \beta \delta^{2}\eta_{1}\omega_{j} & 0\\ 0 & \beta^{2}\gamma^{2} & 0 & 0 & 0 & 0 & 0 & \beta \gamma^{2}\eta_{1}\omega_{j}\\ 0 & 0 & \alpha^{2}\delta^{2} & 0 & 0 & 0 & 0 & \alpha^{2}\delta \eta_{2}\omega_{k}\\ 0 & 0 & 0 & \alpha^{2}\gamma^{2} & 0 & 0 & 0 & 0\\ 0 & 0 & 0 & 0 & 0 & 0 & 0 & 0\\ \beta^{2}\delta \eta_{2}\omega_{k}^{*} & 0 & 0 & 0 & 0 & \beta^{2}\eta_{2}^{2}{\left|\omega_{k}\right|}^{2} & \beta \delta \eta_{1}\eta_{2}\omega_{j}\omega_{k}^{*} & 0\\ \beta \delta^{2}\eta_{1}\omega_{j}^{*} & 0 & 0 & 0 & 0 & \beta \delta \eta_{1}\eta_{2}\omega_{j}^{*}\omega_{k} & \delta^{2}\eta_{1}^{2}{\left|\omega_{j}\right|}^{2} & 0\\ 0 & \beta \gamma^{2}\eta_{1}\omega_{j}^{*} & \alpha^{2}\delta \eta_{2}\omega_{k}^{*} & 0 & 0 & 0 & 0 & \gamma^{2}\eta_{1}^{2}|\omega_{j}{|}^{2}+\alpha^{2}\eta_{2}^{2}{\left|\omega_{k}\right|}^{2} \end{array}\right), \end{array}$$
where $\alpha =\sin\left(r_{j}\right),\beta =\cos\left(r_{j}\right),\gamma =\sin\left(r_{k}\right),\delta =\cos\left(r_{k}\right),\eta_{1}=\cos\left(2r_{j}\right),\eta_{2}=\cos\left(2r_{k}\right)$. For the bipartite subsystem $\rho_{\mathrm{AjI}}$, its matrix form is given by
(24)$$\rho_{\mathrm{AjI}=}\frac{1}{2}\left(\begin{array}{cccc} \beta^{2} & 0 & 0 & \left(2\beta^{3}-\beta \right)\omega_{j}\\ 0 & \alpha^{2} & 0 & 0\\ 0 & 0 & \beta^{2}\eta_{2}^{2}{\left|\omega_{k}\right|}^{2} & 0\\ \left(2\beta^{3}-\beta \right)\omega_{j}^{*} & 0 & 0 & \eta_{1}^{2}|\omega_{j}{|}^{2}+\alpha^{2}\eta_{2}^{2}{\left|\omega_{k}\right|}^{2} \end{array}\right).$$
The matrix $\rho_{\mathrm{AkI}}$ can easily be obtained from $\rho_{\mathrm{AjI}}$ by replacing $r_{j}$ with $r_{k}$ and $\omega_{j}$ with $\omega_{k}$, respectively. In this case, we will use the negativities and von Neumann entropy to show its entanglement properties.

To calculate the negativity, we need to obtain the partial transpose of the density matrix [35]. After this process, if the density matrix has at least one negative eigenvalue, we can say the density matrix is entangled. The negativity is defined as [36,37]
(25)$$N_{\alpha (\beta \gamma )}=\left|\right|\rho_{\alpha (\beta \gamma )}^{T_{\alpha }}\left|\right|-1=2\sum_{i=1}^{N}|\lambda_{M}^{\left(-\right)}{|}^{i},\;\;N_{\alpha \beta }=\left|\right|\rho_{\alpha \beta }^{T_{\alpha }}\left|\right|-1,$$
where $\lambda_{M}^{\left(-\right)}$ are the negative eigenvalues of the matrix *M*.

When we study an entangled quantum system, it is also necessary to study the von Neumann entropy defined as [38],
(26)$$\begin{array}{c} S=-\mathrm{Tr}\left(\rho \log_{2}\rho \right)=-\sum_{i=1}^{n}\lambda^{\left(i\right)}\log_{2}\lambda^{\left(i\right)}, \end{array}$$
where $\lambda^{\left(i\right)}$ denotes the *i*-th nonzero eigenvalue of the density matrix ρ. It should be pointed out that the density matrix is not taken as its partial transpose. Thus, we are able to use it to measure the stability of the studied quantum system.

We illustrate the negativity in Figure 1 and notice that the degree of the entanglement always decreases with the acceleration parameters $r_{j}$ and $r_{k}$, but the degree of entanglement for all of them still exists even in the acceleration limit r→π/4.
(27)$$\begin{array}{c} N_{A(jI)}=N_{jI(A)}=\\ \;\;\;\;\;\;\;\;\;\frac{1}{4}[-2{\left|\omega_{k}\right|}^{2}\cos^{2}\left(r_{j}\right)\cos^{2}\left(2r_{k}\right)+\cos\left(2r_{j}\right)-1\\ \;\;\;\;\;\;\;\;\;+2\{{\left(\sin^{2}\left(r_{j}\right)-{\left|\omega_{k}\right|}^{2}\cos^{2}\left(r_{j}\right)\cos^{2}\left(2r_{k}\right)\right)}^{2}\\ \;\;\;\;\;\;\;\;\;+4|\omega_{j}{{|}^{2}\cos^{2}\left(r_{j}\right)\cos^{2}\left(2r_{j}\right)\}}^{1/2}]. \end{array}$$
Due to the symmetry, one has $N_{A(kI)}=N_{kI(A)}$, which can be easily obtained by exchanging $\omega_{j}$ and $\omega_{k}$ of $N_{A(jI)}=N_{jI(A)}$. As illustrated in Figure 2, we notice that the degree of the entanglement for the negativity ($N_{A(jI)}=N_{A(kI)}$ by exchanging $\omega_{j}↔\omega_{k}$ due to symmetry) always decreases with the acceleration parameters $r_{j}$ ($r_{k}$) but increases with the increasing of the acceleration parameter $r_{k}$ ($r_{j}$). As $\omega_{j}/\omega_{k}$ increases, the negativities increase, too. For convenience, we take $\left\{\omega_{j},\omega_{k}\right\}$ as follows:(28)$$\left\{\omega_{j},\omega_{k}\right\}=\left\{\frac{1}{2},\frac{\sqrt{3}}{2}\right\},\left\{\frac{\sqrt{2}}{2},\frac{\sqrt{2}}{2}\right\},\left\{\frac{\sqrt{3}}{2},\frac{1}{2}\right\}.$$

In particular, it is found that the negativity $N_{A(jI)}$ will disappear when $r_{j}$ exceeds some values, which are proportional to the ratio of the $\omega_{j}/\omega_{k}$.

To calculate the von Neumann entropy, we are going to present eigenvalues of all subsystems and whole systems for the case of the Bell state as follows:(29)$$\begin{array}{c} \lambda_{AjIkI}^{\left(1\right)}=\frac{1}{4}[2\cos^{2}\left(r_{k}\right)\left(|\omega_{j}{|}^{2}\cos^{2}\left(2r_{j}\right)+\cos^{2}\left(r_{j}\right)\right)\\ \;\;\;\;\;\;\;+|\omega_{k}{|}^{2}\cos^{2}\left(r_{j}\right)\left(\cos\left(4r_{k}\right)+1\right)],\\ \lambda_{AjIkI}^{\left(2,3\right)}=\frac{1}{8}\{\mp [4{\left|\omega_{j}\right|}^{4}\cos^{4}\left(2r_{j}\right)\sin^{4}\left(r_{k}\right)+4\cos^{2}\left(2r_{j}\right)\sin^{2}\left(r_{k}\right)\\ \;\;\;\;\;\;\;|\omega_{j}{|}^{2}\left(\cos\left(2r_{j}\right)-\cos\left(2r_{k}\right)+2{\left|\omega_{k}\right|}^{2}\sin^{2}\left(r_{j}\right)\cos^{2}\left(2r_{k}\right)\right)\\ \;\;\;\;\;\;\;+{\left(-\cos\left(2r_{j}\right)2{\left|\omega_{k}\right|}^{2}\sin^{2}\left(r_{j}\right)\cos^{2}\left(2r_{k}\right)+\cos\left(2r_{k}\right)\right)}^{2}{]}^{1/2}\\ \;\;\;\;\;\;\;+2|\omega_{j}{|}^{2}\cos^{2}\left(2r_{j}\right)\sin^{2}\left(r_{k}\right)+2{\left|\omega_{k}\right|}^{2}\sin^{2}\left(r_{j}\right)\cos^{2}\left(2r_{k}\right)\\ \;\;\;\;\;\;\;-\cos\left(2r_{j}\right)\cos\left(2r_{k}\right)+1\},\\ \lambda_{AjIkI}^{\left(4\right)}=\frac{1}{2}\sin^{2}\left(r_{j}\right)\sin^{2}\left(r_{k}\right), \end{array}$$
(30)$$\begin{array}{c} \lambda_{AjI}^{\left(1\right)}=\frac{1}{2}{\left|\omega_{k}\right|}^{2}\cos^{2}\left(r_{j}\right)\cos^{2}\left(2r_{k}\right),\\ \lambda_{AjI}^{\left(2,3\right)}=\frac{1}{8}\{\mp 2[{\left|\omega_{k}\right|}^{2}\sin^{2}\left(2r_{j}\right)\left(-\cos^{2}\left(2r_{k}\right)\right)\\ \;\;+{\left(|\omega_{k}{|}^{2}\sin^{2}\left(r_{j}\right)\cos^{2}\left(2r_{k}\right)+{\left|\omega_{j}\right|}^{2}\cos^{2}\left(2r_{j}\right)+\cos^{2}\left(r_{j}\right)\right)}^{2}{]}^{1/2}\\ \;\;+2\left(|\omega_{k}{|}^{2}\sin^{2}\left(r_{j}\right)\cos^{2}\left(2r_{k}\right)+\cos^{2}\left(r_{j}\right)\right)+|\omega_{j}{|}^{2}\cos\left(4r_{j}\right)+{\left|\omega_{j}\right|}^{2}\},\\ \lambda_{AjI}^{\left(4\right)}=\frac{1}{2}\sin^{2}\left(r_{j}\right), \end{array}$$
(31)$$\begin{array}{c} \lambda_{jIkI}^{\left(1\right)}=\frac{1}{2}\cos^{2}\left(r_{j}\right)\cos^{2}\left(r_{k}\right),\\ \lambda_{jIkI}^{\left(2,3\right)}=\frac{1}{8}\{\mp [4{\left|\omega_{j}\right|}^{4}\cos^{4}\left(2r_{j}\right)\cos^{4}\left(r_{k}\right)\\ \;\;\;\;+{\left(\cos\left(2r_{j}\right)2{\left|\omega_{k}\right|}^{2}\cos^{2}\left(r_{j}\right)\cos^{2}\left(2r_{k}\right)-\cos\left(2r_{k}\right)\right)}^{2}\\ \;\;\;\;+4|\omega_{j}{|}^{2}\cos^{2}\left(2r_{j}\right)\cos^{2}\left(r_{k}\right)\\ \;\;\;\;\left(-\cos\left(2r_{j}\right)2{\left|\omega_{k}\right|}^{2}\cos^{2}\left(r_{j}\right)\cos^{2}\left(2r_{k}\right)+\cos\left(2r_{k}\right)\right){]}^{1/2}\\ \;\;\;\;+2|\omega_{j}{|}^{2}\cos^{2}\left(2r_{j}\right)\cos^{2}\left(r_{k}\right)+2{\left|\omega_{k}\right|}^{2}\cos^{2}\left(r_{j}\right)\cos^{2}\left(2r_{k}\right)\\ \;\;\;\;-\cos\left(2r_{j}\right)\cos\left(2r_{k}\right)+1\},\\ \lambda_{jIkI}^{\left(4\right)}=\frac{1}{2}\left[\right|\omega_{j}{|}^{2}\cos^{2}\left(2r_{j}\right)\sin^{2}\left(r_{k}\right)+{\left|\omega_{k}\right|}^{2}\sin^{2}\left(r_{j}\right)\cos^{2}\left(2r_{k}\right)\\ \;\;\;\;\;\;\;+\sin^{2}\left(r_{j}\right)\sin^{2}\left(r_{k}\right)], \end{array}$$
where superscripts 2 and 3 in $\lambda_{AjIkI}^{\left(2,3\right)}$ and $\lambda_{jIkI}^{\left(2,3\right)}$ correspond to the signs “−” and “+” of the expressions, respectively.

Likewise, the $\lambda_{AkI}^{\left(1,2,3,4\right)}$ can be obtained directly from $\lambda_{AjI}^{\left(1,2,3,4\right)}$ by exchanging $\omega_{k}↔\omega_{j}$. As shown in Figure 3, the entropy $S_{AjIkI}$ and $S_{jIkI}$ increase with the increasing acceleration parameters $r_{j}$ and $r_{k}$. However, $S_{AjI}$ ($S_{AkI}$) increases with the acceleration parameters $r_{j}$ but decreases with the acceleration parameters $r_{k}$. This is because $S_{AjI}$, which is only concerned with the observer Bob confined in region I is mainly from the contribution of the $\omega_{j}$. To satisfy the constraint $|\omega_{j}{|}^{2}+{\left|\omega_{k}\right|}^{2}=1$, the entropy $S_{AjI}$ increases with the acceleration parameter $r_{j}$ but has to be decreased with parameter $r_{k}$.

We are now in the position to study the GHZ state, which has the following form in an inertial frame
(32)$$|\phi \rangle =\frac{1}{\sqrt{2}}\left(|0_{\hat{A}}0_{\hat{B}}0_{\hat{C}}\rangle +|1_{\hat{A}}1_{\hat{B}}1_{\hat{C}}\rangle \right).$$
Following a similar process to the Bell state studied above, we obtain the matrix form by tracing over the inaccessible Rindler modes in region II
(33)$$\begin{array}{c} \rho_{ABC_{jI}C_{kI}}=\\ \frac{1}{2}\left(\begin{array}{cccccccccccccccc} \beta^{2}\delta^{2} & 0 & 0 & 0 & 0 & 0 & 0 & 0 & 0 & 0 & 0 & 0 & 0 & \beta^{2}\delta \eta_{2}\omega_{k} & \beta \delta^{2}\eta_{1}\omega_{j} & 0\\ 0 & \beta^{2}\gamma^{2} & 0 & 0 & 0 & 0 & 0 & 0 & 0 & 0 & 0 & 0 & 0 & 0 & 0 & \beta \gamma^{2}\eta_{1}\omega_{j}\\ 0 & 0 & \alpha^{2}\delta^{2} & 0 & 0 & 0 & 0 & 0 & 0 & 0 & 0 & 0 & 0 & 0 & 0 & \alpha^{2}\delta \eta_{2}\omega_{k}\\ 0 & 0 & 0 & \alpha^{2}\gamma^{2} & 0 & 0 & 0 & 0 & 0 & 0 & 0 & 0 & 0 & 0 & 0 & 0\\ 0 & 0 & 0 & 0 & 0 & 0 & 0 & 0 & 0 & 0 & 0 & 0 & 0 & 0 & 0 & 0\\ 0 & 0 & 0 & 0 & 0 & 0 & 0 & 0 & 0 & 0 & 0 & 0 & 0 & 0 & 0 & 0\\ 0 & 0 & 0 & 0 & 0 & 0 & 0 & 0 & 0 & 0 & 0 & 0 & 0 & 0 & 0 & 0\\ 0 & 0 & 0 & 0 & 0 & 0 & 0 & 0 & 0 & 0 & 0 & 0 & 0 & 0 & 0 & 0\\ 0 & 0 & 0 & 0 & 0 & 0 & 0 & 0 & 0 & 0 & 0 & 0 & 0 & 0 & 0 & 0\\ 0 & 0 & 0 & 0 & 0 & 0 & 0 & 0 & 0 & 0 & 0 & 0 & 0 & 0 & 0 & 0\\ 0 & 0 & 0 & 0 & 0 & 0 & 0 & 0 & 0 & 0 & 0 & 0 & 0 & 0 & 0 & 0\\ 0 & 0 & 0 & 0 & 0 & 0 & 0 & 0 & 0 & 0 & 0 & 0 & 0 & 0 & 0 & 0\\ 0 & 0 & 0 & 0 & 0 & 0 & 0 & 0 & 0 & 0 & 0 & 0 & 0 & 0 & 0 & 0\\ \beta^{2}\delta \eta_{2}\omega_{k}^{*} & 0 & 0 & 0 & 0 & 0 & 0 & 0 & 0 & 0 & 0 & 0 & 0 & \beta^{2}\eta_{2}^{2}{\left|\omega_{k}\right|}^{2} & \beta \delta \eta_{1}\eta_{2}\omega_{j}\omega_{k}^{*} & 0\\ \beta \delta^{2}\eta_{1}\omega_{j}^{*} & 0 & 0 & 0 & 0 & 0 & 0 & 0 & 0 & 0 & 0 & 0 & 0 & \beta \delta \eta_{1}\eta_{2}\omega_{j}^{*}\omega_{k} & \delta^{2}\eta_{1}^{2}{\left|\omega_{j}\right|}^{2} & 0\\ 0 & \beta \gamma^{2}\eta_{1}\omega_{j}^{*} & \alpha^{2}\delta \eta_{2}\omega_{k}^{*} & 0 & 0 & 0 & 0 & 0 & 0 & 0 & 0 & 0 & 0 & 0 & 0 & \gamma^{2}\eta_{1}^{2}|\omega_{j}{|}^{2}+\alpha^{2}\eta_{2}^{2}{\left|\omega_{k}\right|}^{2} \end{array}\right), \end{array}$$
where $\alpha =\sin\left(r_{j}\right),\beta =\cos\left(r_{j}\right),\gamma =\sin\left(r_{k}\right),\delta =\cos\left(r_{k}\right),\eta_{1}=\cos\left(2r_{j}\right),\eta_{2}=\cos\left(2r_{k}\right)$ as defined above.

We first illustrate the negativity for this case. Similar to the Bell state case, as shown in Figure 4, we also notice that the degree of the entanglement always decreases with the acceleration parameters $r_{j}$ and $r_{k}$, while the degree of entanglement for all them still exists even in the acceleration limit r→π/4. The negative eigenvalues of the negativities $N_{A(BC_{jI}C_{kI})}$, $N_{B(AC_{jI}C_{kI})}$, $N_{C_{jI}C_{kI}\left(AB\right)}$ and $N_{AB(C_{jI}C_{kI})}$ are written out explicitly in Appendix A and Appendix B. If we only consider the subsystem, we can partially track Alice, Bob, Charlie *j*I or Charlie *k*I independently. Based on Equation (Equation 25), we are able to calculate the corresponding negativites. For this GHZ state, all 1-1 tangle negativities are equal to 0.

This means that in the subsystems, the entanglement does not exist any more.

In this part, we use the von Neumann Entropy to measure the degree of the stability of the studied quantum state. In Figure 5 and Figure 6, we can observe the behavior of the von Neumann entropy, and here, we present all the eigenvalues of the whole system and all subsystems.
(34)$$\begin{array}{c} \lambda_{AB}^{\left(1\right)}=\frac{1}{2},\\ \lambda_{AB}^{\left(2\right)}=\frac{1}{4}\left(|\omega_{j}{|}^{2}\cos\left(4r_{j}\right)+|\omega_{j}{|}^{2}+|\omega_{k}{|}^{2}\cos\left(4r_{k}\right)+{\left|\omega_{k}\right|}^{2}\right) \end{array}$$
(35)$$\begin{array}{c} \lambda_{AC_{jI}}^{\left(1\right)}=\frac{1}{2}\cos^{2}\left(r_{j}\right),\\ \lambda_{AC_{jI}}^{\left(2\right)}=\frac{1}{2}\sin^{2}\left(r_{j}\right),\\ \lambda_{AC_{jI}}^{\left(3\right)}=\frac{1}{2}{\left|\omega_{k}\right|}^{2}\cos^{2}\left(r_{j}\right)\cos^{2}\left(2r_{k}\right),\\ \lambda_{AC_{jI}}^{\left(4\right)}=\frac{1}{2}\left(|\omega_{k}{|}^{2}\sin^{2}\left(r_{j}\right)\cos^{2}\left(2r_{k}\right)+{\left|\omega_{j}\right|}^{2}\cos^{2}\left(2r_{j}\right)\right), \end{array}$$
(36)$$\begin{array}{c} \lambda_{ABC_{jI}}^{\left(1\right)}=\frac{1}{2}\sin^{2}\left(r_{j}\right),\\ \lambda_{ABC_{jI}}^{\left(2\right)}=\frac{1}{2}{\left|\omega_{k}\right|}^{2}\cos^{2}\left(r_{j}\right)\cos^{2}\left(2r_{k}\right),\\ \lambda_{ABC_{jI}}^{\left(3,4\right)}=\frac{1}{8}\left\{\mp 2[(\right|\omega_{k}{|}^{2}\sin^{2}\left(r_{j}\right)\cos^{2}\left(2r_{k}\right)+{\left|\omega_{j}\right|}^{2}\cos^{2}\left(2r_{j}\right),\\ \;\;\;\;+\cos^{2}\left(r_{j}\right){)}^{2}-{\left|\omega_{k}\right|}^{2}\sin^{2}\left(2r_{j}\right)\cos^{2}\left(2r_{k}\right){]}^{1/2}\\ \;\;\;\;\;2\left(|\omega_{k}{|}^{2}\sin^{2}\left(r_{j}\right)\cos^{2}\left(2r_{k}\right)+\cos^{2}\left(r_{j}\right)\right)\\ \;\;\;\;+|\omega_{j}{|}^{2}\left(\cos\left(4r_{j}\right)+1\right)\}, \end{array}$$
(37)$$\begin{array}{c} \lambda_{C_{jI}C_{kI}}^{\left(1\right)}=\frac{1}{2}\cos^{2}\left(r_{j}\right)\cos^{2}\left(r_{k}\right),\\ \lambda_{C_{jI}C_{kI}}^{\left(2\right)}=\frac{1}{2}[{\left|\omega_{j}\right|}^{2}\cos^{2}\left(2r_{j}\right)\sin^{2}\left(r_{k}\right)\\ \;\;\;\;+|\omega_{k}{|}^{2}\sin^{2}\left(r_{j}\right)\cos^{2}\left(2r_{k}\right)+\sin^{2}\left(r_{j}\right)\sin^{2}\left(r_{k}\right)],\\ \lambda_{C_{jI}C_{kI}}^{\left(3,4\right)}=\frac{1}{8}\{\mp [4{\left|\omega_{j}\right|}^{4}\cos^{4}\left(2r_{j}\right)\cos^{4}\left(r_{k}\right)\\ \;\;\;\;+{\left(\cos\left(2r_{j}\right)2{\left|\omega_{k}\right|}^{2}\cos^{2}\left(r_{j}\right)\cos^{2}\left(2r_{k}\right)-\cos\left(2r_{k}\right)\right)}^{2}\\ \;\;\;\;+4\cos^{2}\left(2r_{j}\right)\cos^{2}\left(r_{k}\right){\left|\omega_{j}\right|}^{2}\\ \;\;\;\;\left(-\cos\left(2r_{j}\right)+2{\left|\omega_{k}\right|}^{2}\cos^{2}\left(r_{j}\right)\cos^{2}\left(2r_{k}\right)+\cos\left(2r_{k}\right)\right){]}^{1/2}\\ \;\;\;\;+2|\omega_{j}{|}^{2}\cos^{2}\left(2r_{j}\right)\cos^{2}\left(r_{k}\right)+2{\left|\omega_{k}\right|}^{2}\cos^{2}\left(r_{j}\right)\cos^{2}\left(2r_{k}\right)\\ \;\;\;\;-\cos\left(2r_{j}\right)\cos\left(2r_{k}\right)+1\}, \end{array}$$
(38)$$\begin{array}{c} \lambda_{AC_{jI}C_{kI}}^{\left(1\right)}=\frac{1}{2}\cos^{2}\left(r_{j}\right)\cos^{2}\left(r_{k}\right),\\ \lambda_{AC_{jI}C_{kI}}^{\left(2\right)}=\frac{1}{2}\sin^{2}\left(r_{j}\right)\cos^{2}\left(r_{k}\right),\\ \lambda_{AC_{jI}C_{kI}}^{\left(3\right)}=\frac{1}{2}\cos^{2}\left(r_{j}\right)\sin^{2}\left(r_{k}\right),\\ \lambda_{AC_{jI}C_{kI}}^{\left(4\right)}=\frac{1}{2}\sin^{2}\left(r_{j}\right)\sin^{2}\left(r_{k}\right),\\ \lambda_{AC_{jI}C_{kI}}^{\left(5\right)}=\frac{1}{2}[{\left|\omega_{j}\right|}^{2}\cos^{2}\left(2r_{j}\right)\cos^{2}\left(r_{k}\right)\\ \;\;\;\;+|\omega_{k}{|}^{2}\cos^{2}\left(r_{j}\right)\cos^{2}\left(2r_{k}\right)],\\ \lambda_{AC_{jI}C_{kI}}^{\left(6\right)}=\frac{1}{2}[{\left|\omega_{j}\right|}^{2}\cos^{2}\left(2r_{j}\right)\sin^{2}\left(r_{k}\right)\\ \;\;\;\;+|\omega_{k}{|}^{2}\sin^{2}\left(r_{j}\right)\cos^{2}\left(2r_{k}\right)], \end{array}$$
(39)$$\begin{array}{c} \lambda_{ABC_{jI}C_{kI}}^{\left(1\right)}=\frac{1}{2}\sin^{2}\left(r_{k}\right)\sin^{2}\left(r_{j}\right),\\ \lambda_{ABC_{jI}C_{kI}}^{\left(2\right)}=\frac{1}{4}\{2\cos^{2}\left(r_{k}\right)\left(|\omega_{j}{|}^{2}\cos^{2}\left(2r_{j}\right)+\cos^{2}\left(r_{j}\right)\right)\\ \;\;\;\;+|\omega_{k}{|}^{2}\cos^{2}\left(r_{j}\right)\left(\cos\left(4r_{k}\right)+1\right)\},\\ \lambda_{ABC_{jI}C_{kI}}^{\left(3,4\right)}=\frac{1}{8}\{\mp [(2{\left|\omega_{k}\right|}^{2}\sin^{2}\left(r_{j}\right)\cos^{2}\left(2r_{k}\right)\\ \;\;\;\;-\cos\left(2r_{j}\right)+\cos\left(2r_{k}\right){)}^{2}+4{\left|\omega_{j}\right|}^{4}\cos^{4}\left(2r_{j}\right)\sin^{4}\left(r_{k}\right)\\ \;\;\;\;+4\cos^{2}\left(2r_{j}\right)\sin^{2}\left(r_{k}\right){\left|\omega_{j}\right|}^{2}(\cos\left(2r_{j}\right)-\cos\left(2r_{k}\right)\\ \;\;\;\;+2|\omega_{k}{|}^{2}\sin^{2}\left(r_{j}\right)\cos^{2}\left(2r_{k}\right)){]}^{1/2}\\ \;\;\;\;+2|\omega_{j}{|}^{2}\cos^{2}\left(2r_{j}\right)\sin^{2}\left(r_{k}\right)\\ \;\;\;\;+2|\omega_{k}{|}^{2}\sin^{2}\left(r_{j}\right)\cos^{2}\left(2r_{k}\right)\\ \;\;\;\;-\cos\left(2r_{j}\right)\cos\left(2r_{k}\right)+1\}. \end{array}$$
It should be noted that $\lambda_{AC_{kI}}$ and $\lambda_{ABC_{kI}}$ can easily be obtained from $\lambda_{AC_{jI}}$ and $\lambda_{ABC_{jI}}$ by exchanging $r_{j}↔r_{k}$ and $\omega_{j}↔\omega_{k}$.

**Figure 5 entropy-24-01011-f005:**
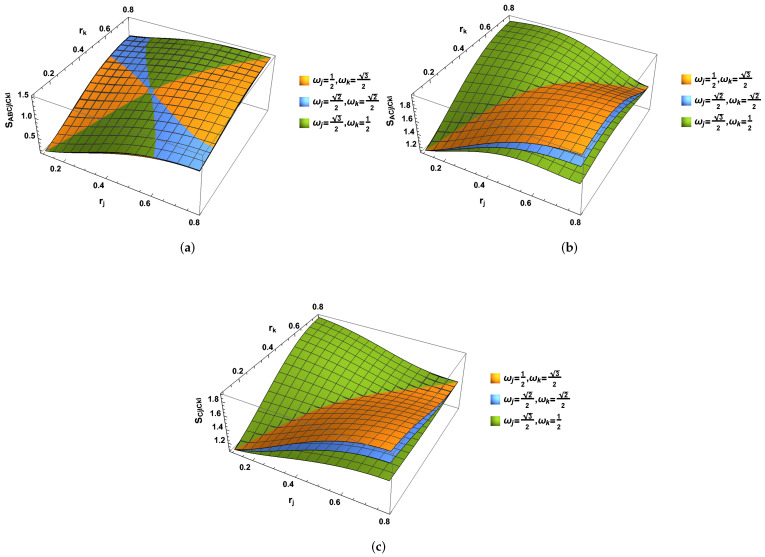
(Color online) The von Neumann entropies $S_{ABC_{jI}C_{kI}}$, $S_{AC_{jI}C_{kI}}$ and $S_{C_{jI}C_{kI}}$ as the functions of both acceleration parameters $r_{j}$ and $r_{k}$.

**Figure 6 entropy-24-01011-f006:**
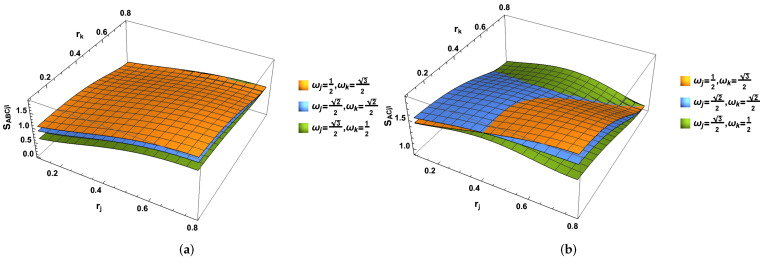
(Color online) The von Neumann Entropies $S_{ABCjI}$ (or $S_{ABCkI}$) and $S_{ACjI}$ (or $S_{ACkI}$) as the functions of both acceleration parameters $r_{j}$ or $r_{k}$.

In Figure 5, we can see that the entropy $S_{ABC_{jI}C_{kI}}$, $S_{AC_{jI}C_{kI}}$ and $S_{C_{jI}C_{kI}}$ increase with the increasing acceleration parameter $r_{j}$ and $r_{k}$. If we only consider the *j* mode (*k* mode), we can partially trace the *k* mode (*j* mode). It is seen that von Neumann entropies and $S_{ABCjI}$ increase with the increasing $r_{j}$. However, von Neumann entropies $S_{AjI}$ in the Bell case and $S_{ACjI}$ in GHZ state first increase and then decrease with the increasing $r_{j}$. All of them $S_{ACjI}$, $S_{ABCjI}$ and $S_{ACjI}$, always decrease with the acceleration parameter $r_{k}$.

## 4. Concluding Remarks

In this work, we have first presented analytical expressions of the Minkowski states |0〉 and |1〉 by taking different accelerating observer frames into account. We used the transformation to test the degree of the Bell state’s entanglement by computing the negativity and the von Neumann entropy. For negativity, we can see that for the whole system, the negativity is always positive. However, for the subsystems, if we only consider one mode, the entanglement of the state depends on the radio $\omega_{j}/\omega_{k}$. For the von Neumann entropy of the whole system, we have observed that the entropy increases as the increasing $r_{j}$ and $r_{k}$ in the entangled system. However, it is shown from the von Neumann entropy of the subsystem, e.g., $S_{ABCjI}$ and $S_{ACjI}$, that the former increases with the increasing $r_{j}$, but the latter first increases and then decreases with the increasing $r_{j}$. Both $S_{ABCjI}$ and $S_{ACjI}$ always decrease with the acceleration parameter $r_{k}$.

## Figures and Tables

**Figure 1 entropy-24-01011-f001:**
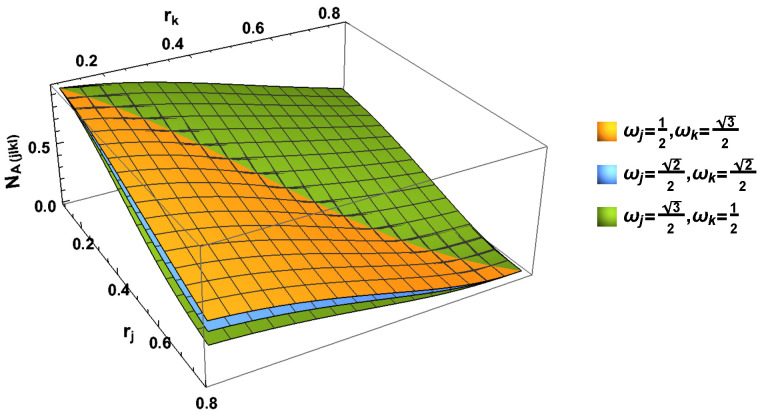
(Color online) Negativity $N_{A(jIkI)}$ plotted as the function of acceleration parameters $r_{j}$ and $r_{k}$.

**Figure 2 entropy-24-01011-f002:**
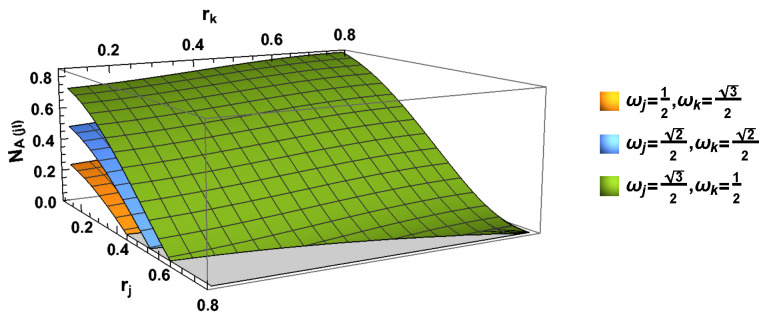
(Color online) The negativity $N_{A(jI)}$ (equally $N_{A(kI)}$) as the functions of acceleration parameters $r_{j}$ or $r_{k}$.

**Figure 3 entropy-24-01011-f003:**
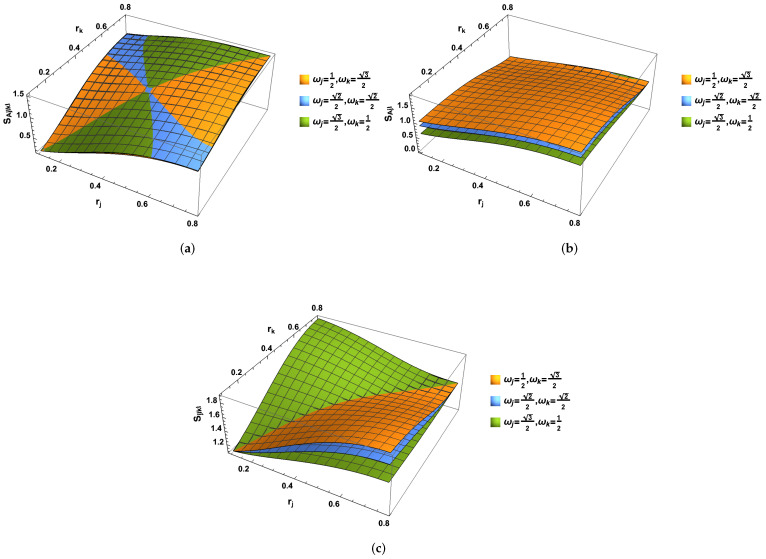
(Color online) The von Neumann entropies (**a**) $S_{AjIkI}$, (**b**) $S_{AjI}$ ($S_{AkI}$) and (**c**) $S_{kIjI}$ as the functions of both acceleration parameters $r_{j}$ and $r_{k}$. It is found that the entropy $S_{AjIkI}$ and $S_{jIkI}$ increase with the increasing acceleration parameters $r_{j}$ and $r_{k}$. However, the variation of $S_{AjI}$ ($S_{AkI}$) with respect to them is different from $S_{AjIkI}$ and $S_{jIkI}$. We notice that $S_{AjI}$ ($S_{AkI}$) increases with the acceleration parameters $r_{j}$, whereas it decreases with the acceleration parameters $r_{k}$.

**Figure 4 entropy-24-01011-f004:**
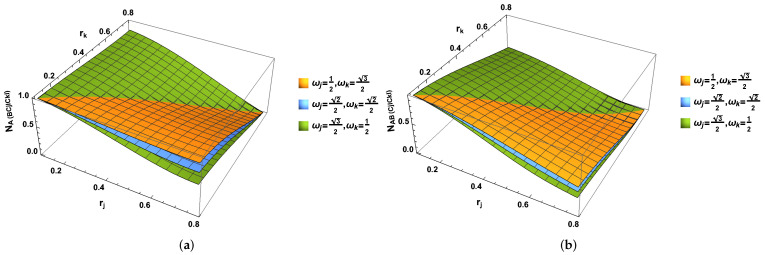
(Color online) The negativity $N_{A(BC_{jI}C_{kI})}$(or $N_{B(AC_{jI}C_{kI})}$) and $N_{AB(C_{jI}C_{kI})}$ (or $N_{C_{jI}C_{kI}\left(AB\right)}$) as the functions of both acceleration parameters $r_{j}$ and $r_{k}$.

## Data Availability

The study did not report any data.

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
