# Peer review of "Entanglement Property of Tripartite GHZ State in Different Accelerating Observer Framesâ€"

_entropy, 2022, doi:10.3390/e24081011_

Round 1
Reviewer 1 Report
This paper considers measures of entanglement for elements of a Dirac field Fock space in a uniformly accelerating reference frame. Prior work has considered this problem for a pair of particles in the single-mode approximation. The authors claim to expand upon this work by considering the two-mode approximation. It should be clarified, however, that what the authors are doing is still the single-mode approximation, albeit for multiple modes, each with a potentially different accelerating observer frame.
The paper contains a technical error in Sec. III that impacts the remaining sections of the paper. The error will need to be corrected and the remainder of the paper will need to be revised accordingly. Specifically, after carefully examining their work I found Eqn. (32) to be in error. The authors appear to have incorrectly mistook expressions of form cos^2(r) - sin^2(r) = cos(2r) to be cos^2(r) + sin^(r) = 1. Although correcting this error is straightforward, its impact on later sections is unknown.
There are several other issues that should be addressed in a revised submission. The derivation of Eqn. (15) is identical, almost verbatim, to that of Alsing 2006 (Ref. [28]). Although Alsing 2006 is cited in the introduction, it is not cited specifically here. The results, with the relevant preliminary definitions and assumptions, could be easily summarized instead.
In Sec. III the two-mode approximation is introduced as a superposition of two annihilation operators on modes j and k, acting in Region I. The complex coefficients of this superposition are given as omega_j and omega_k, respectively. This is a poor choice of notation, as it can easily be confused with the mode frequencies defined in Eqn. (2). The normalization condition for these parameters is not specified until near the end of the section, in Eqn. (30), and there the authors fail to note that it is the squared magnitudes, not merely the squares, that must sum to unity.
The notation, with its many subscripts and superscripts, is confusing, although this is largely, and regrettably, in keeping with past work on this subject. The capital roman numerals I and II should be in roman font and not treated as italicized variables. I am also not a fan of constructions such as c_k^{I,II} to indicate the two distinct operators c_k^\mathrm{I} and c_k^\mathrm{II}, a practice is, again, all too common. Similarly, while putting subscripts on literals, such as 0_k, is sufficiently unambiguous so as not to cause confusion, constructions such as that in Eqn. (12), where n_k and n_{-k} represent that same value of n for different kets is somewhat more confusing. Again, this is in keeping with common practice with past work. I'm hoping the authors can do better.
The derivation in Sec. III is needlessly complicated by extraneous material and "throw away" notation, such as \bar{omega}_k, that is never defined and never used. The phase phi is introduced initially with no subscript, then later with a subscript applied. The "derivation" of Eqn. (26) follows from the convention used in Eqn. (15), one of several that may be adopted. I suggest the authors reconsider the end result of this section and tailor to derivation to cleanly provide this result with all assumptions stated up front.
Reviewer 2 Report
This work considers some relativistic aspects of quantum entanglement in the two-mode approximation. The authors provide a detailed derivation starting from the single-mode case and next generalizing it through an appropriate squeezing transformation. Finally, they apply the results to the case of entangled fermionic fields where they quantify the satiability using the quantum (von Neumann) entropy. The paper is sound and it is worthy of publication. What I’m missing here is the physical interpretation of the results. Also, what possible application in real life these results could find besides their purely academic significance. Also, it seems that Refs.4 and 34 are the same...
Round 2
Reviewer 1 Report
The abstract contains variables that are undefined. No one reading just the abstract could understand what was done and what conclusions were drawn.
The authors still refer to their method as a two-mode approximation, but it is the single-mode approximation applied to two different modes, each associated with different uniformly accelerating reference frames.
Reviewer 2 Report
The authors have responded adequately. Accept as is.
Author Response
Dear Reviewer: Many thanks for your kind recommendation.